# Energy Restriction Enhances Adult Hippocampal Neurogenesis-Associated Memory after Four Weeks in an Adult Human Population with Central Obesity; a Randomized Controlled Trial

**DOI:** 10.3390/nu12030638

**Published:** 2020-02-28

**Authors:** Curie Kim, Ana Margarida Pinto, Claire Bordoli, Luke Patrick Buckner, Polly Charlotte Kaplan, Ines Maria del Arenal, Emma Jane Jeffcock, Wendy L Hall, Sandrine Thuret

**Affiliations:** 1Department of Basic and Clinical Neuroscience, Institute of Psychiatry, Psychology and Neuroscience, King’s College London, 125 Coldharbour Lane, London SE5 9NU, UK; 2Department of Nutritional Sciences, School of Life Course Sciences, Faculty of Life Sciences and Medicine, King’s College London, Franklin-Wilkins Building, 150 Stamford Street, London SE1 9NH, UK; 3Department of Neurology, University Hospital Carl Gustav Carus, Technische Universität Dresden, 01307 Dresden, Germany

**Keywords:** adult hippocampal neurogenesis, energy restriction, recognition memory, pattern separation, intermittent fasting, ageing, cognition, randomized controlled trial

## Abstract

Adult neurogenesis, the generation of new neurons throughout life, occurs in the subventricular zone of the dentate gyrus in the human hippocampal formation. It has been shown in rodents that adult hippocampal neurogenesis is needed for pattern separation, the ability to differentially encode small changes derived from similar inputs, and recognition memory, as well as the ability to recognize previously encountered stimuli. Improved hippocampus-dependent cognition and cellular readouts of adult hippocampal neurogenesis have been reported in daily energy restricted and intermittent fasting adult mice. Evidence that nutrition can significantly affect brain structure and function is increasing substantially. This randomized intervention study investigated the effects of intermittent and continuous energy restriction on human hippocampal neurogenesis-related cognition, which has not been reported previously. Pattern separation and recognition memory were measured in 43 individuals with central obesity aged 35–75 years, before and after a four-week dietary intervention using the mnemonic similarity task. Both groups significantly improved pattern separation (*P* = 0.0005), but only the intermittent energy restriction group had a significant deterioration in recognition memory. There were no significant differences in cognitive improvement between the two diets. This is the first human study to investigate the association between energy restriction with neurogenesis-associated cognitive function. Energy restriction may enhance hippocampus-dependent memory and could benefit those in an ageing population with declining cognition. This study was registered on ClinicalTrials.gov (NCT02679989) on 11 February 2016.

## 1. Introduction

It is widely accepted that diet can significantly affect brain structure and function [1]. Evidence to suggest that the quality of diet and weight having a significant impact on cognition has increased substantially. For example, mice on a high fat diet display increased anxiety-like behavior and impaired learning [2,3]. Similar results can be seen in human populations. A cross-sectional study of 428 children concluded that poor diet quality is associated with worse cognition [4]. Whilst on the other end of the ageing spectrum, good nutritional status may delay functional decline in the elderly [5]. 

Human adult hippocampal neurogenesis (AHN), the postnatal process by which new neurons are generated from neural stem cells, has so far been shown to occur in a unique neurogenic niche within the dentate gyrus of the hippocampal formation throughout life, although this remains widely discussed [6,7]. 

Human AHN was first demonstrated in the late 90s by Eriksson et al., showing evidence of newly generated neurons in the dentate gyrus into late adulthood [8]. Retrospective birth-dating using atmospheric ^14^C levels derived from nuclear bomb testing during the cold war gave further evidence of human AHN with a turnover rate of approximately 700 new neurons a day [9]. However, a recent controversial study examining samples of post-mortem hippocampal tissue across the lifespan found no evidence of cell proliferation in the dentate gyrus past adolescence [10]. Nevertheless, contrasting studies have since demonstrated the presence of neural progenitors and neurons at various stages of maturity up to the eighth and ninth decade of life [11,12]. A growing body of evidence from animal studies suggests that AHN has several physiological functions including hippocampus-dependent memory encoding and mood regulation [13,14,15]. This study will focus on the impact of AHN on learning and memory.

Pattern separation and recognition memory are both hippocampus-dependent forms of memory, specifically the dorsal hippocampal formation, which have been demonstrated to be modulated by AHN in rodents [16,17]. Pattern separation is the process by which similar patterns of neuronal inputs are transformed into direct neuronal representations in order to discriminate between highly similar stimuli in human hippocampus-dependent tasks [18]. Functional magnetic resonance imaging in conjunction with pattern separation tasks show that in particular, the CA3, which receives input from the dentate gyrus, is strongly activated [19,20]. Considering the hippocampal formation is a neurogenic niche, it is highly possible that pattern separation performance could be mediated by AHN. This has been demonstrated in rodent studies. Decreased AHN in mice impairs performance in spatial pattern separation tasks, whilst on the other hand, genetic expansion of adult-born neuron populations gives rise to improved pattern separation despite normal object recognition, spatial learning, contextual fear conditioning, and extinction learning [21,22]. It has been indicated that these adult-born neurons may modulate plasticity within the dentate gyrus to facilitate these hippocampus-dependent memory functions [23]. Recognition memory is the ability to judge and recognize an item that has been previously encountered by remembering specific contextual details about a prior learning episode [24]. Similar to pattern separation, the hippocampal formation also has a significant role in recognition memory, although within a much wider network of connections [25]. Temporary or permanent lesions of the whole hippocampal formation and the dorsal hippocampus have been shown to impair recognition memory performance in object recognition tasks in both rodents and primates [26,27]. Furthermore, AHN-enhancing environmental stimuli such as sexual activity and exercise have been shown to increase newly generated neurons and alleviate impairments in recognition memory in rodents [28,29].

Rodent studies have repeatedly shown that AHN can be modulated by nutritional cues including dietary factors such as energy intake and meal frequency [30]. Amongst the multiple aspects of dietary intake, energy restriction in the absence of malnutrition has been attributed to many beneficial effects on brain function, and AHN may be one of the mechanisms behind the positive relationship between energy restriction and cognition [31]. There are two commonly used methods of energy restriction: 1) continuous energy restriction (CER), i.e. a maintained reduced daily calorie intake, and 2) intermittent energy restriction (IER), i.e. drastic reductions in calorie intake for varying periods of time. In human studies, the most common IER approaches include alternate day fasting and the 5:2 pattern (five days of unrestricted eating combined with two days of severe energy restriction each week) [32]. It has become clear that there is a strong relationship between AHN, energy restriction and hippocampus-dependent memory in rodents [33]. Moreover, there is emerging evidence of a positive relationship between energy restriction, human health, and cognition [34]. The potential impact of energy restriction on AHN-dependent cognition in a human population is not yet clear and may eventually help to elucidate the functional role of neurogenesis. Therefore, as a first step towards this long-term goal, we evaluated the impact of energy restriction on a hippocampus-dependent cognitive test measuring pattern separation and recognition memory by carrying out a randomized controlled dietary intervention in adults randomly assigned either CER or IER. We hypothesized that both methods of energy restriction will result in improved memory performance, but that IER would have a greater effect. To the best of our knowledge, there are currently no other studies that have investigated the effects of energy restriction on pattern separation and recognition memory in human participants, and this study aims to explore that prospect.

## 2. Materials and Methods 

### 2.1. Ethical Standards

This study was approved by the King’s College London Research Ethics Committee (ref: HR-15/16-2179) and registered at clinicaltrials.org (NCT02679989). The study was conducted in accordance with the Declaration of Helsinki. All subjects gave written informed consent before any procedures were carried out and received a remuneration for taking part.

### 2.2. Participant Selection

Recruitment was carried out through social media, London-wide newspaper advertisements, and internal circulars at King’s College London and King’s Health Partners. Subjects were required to be healthy, non-smoking men and women aged between 35 and 75 years old with a waist circumference of greater than 102 cm in men and 88 cm in women with a Europid, Black African and Caribbean, or other ethnic background, or greater than 80 cm or 90 cm in women and men, respectively, identifying as Asian. The exclusion criteria included a medical history of myocardial infarction; cerebrovascular accident; angina; liver or kidney disease; diabetes mellitus; chronic gastrointestinal disorder; cancer in the previous five years; uncontrolled epilepsy or depression; bariatric or any other major surgery; pregnant, lactating women or those planning pregnancy; those who participated in any weight management drug trial in the previous three months; those taking medication to lower their weight; a history of excess alcohol intake or substance abuse.

### 2.3. Study Protocol

The Met-IER study (Metabolic effects of Intermittent Energy Restriction) was a randomized controlled dietary intervention trial carried out at King’s College London, United Kingdom between February and July 2016. It was primarily designed and conducted by the Diet and Cardiometabolic Health research group in the Department of Nutritional Sciences of King’s College London to investigate the cardio-metabolic effects of short-term IER (5:2 pattern) and CER in centrally obese men and women [35]. Potential participants attended a screening visit at which anthropometric measurements and blood pressure measurements were taken. A fasting blood sample was collected for a full lipid profile, glucose, liver function, and haematology assessment. 

Eligible participants were randomly allocated to IER or CER by minimization using MinimPy 0.3 (Copyright © 2020 Mahmoud Saghael, http://minimpy.sourceforge.net) with sex, BMI, ethnicity and waist circumference included as minimization factors. Participants attended study visits in the fasting state for measurements of body composition, cognitive testing with the Mnemonic Similarity Task (MST) and a blood sample. Participants also underwent several other procedures for various cardiometabolic measurements which are described elsewhere [35]. All endpoint measurements were taken in duplicate after four to five weeks of the dietary intervention. One endpoint measurement was taken following the two days of a very low-calorie diet for the IER group and another after a minimum two days of normal eating. This allowed the acute effects of fasting within the IER group to be investigated.

The dietary interventions were designed to reduce weekly energy intake by 3500 kcal (14.64 MJ) relative to estimated total energy expenditure, which was calculated from individual resting metabolic rates measured by indirect calorimetry and physical activity estimates using the short form international physical activity questionnaire. Participants randomized to CER were advised by a study student dietitian to consume a nutritionally balanced Mediterranean-style diet with a daily 500 kcal (2.09 MJ) deficit. Participants in the IER group were asked to consume 600 kcal (2.51 MJ) for two consecutive days each week using meal replacement food packs supplied by LighterLife UK Ltd (Harlow, UK). The packs had an average composition of 38% carbohydrate, 36% protein, and 26% fat, and included 100% recommended daily allowance of vitamins and minerals. The fasting alternated with a nutritionally balanced Mediterranean-style diet for the remaining five days. The participants were asked to consume the food packs within 12 hours and have a minimum 12 hours overnight fast. Both interventions were energy-matched with a target to achieve the same weekly energy deficit. In order to maximize compliance, all participants received follow up phone calls and attended one-hour group support sessions.

### 2.4. Mnemonic Similarity Task

The mnemonic similarity task was developed as a method to quantitively measure pattern separation and recognition memory performance in humans [18,36]. It is a 2-phase test involving a series of colour photographs of everyday objects on a white background. The first part of the test is an encoding phase in which subjects were instructed to indicate whether they associated the image shown to them with “indoors” or “outdoors” via a labelled button press (64 items total, two seconds each, 0.5 seconds inter-stimuli interval). Immediately afterwards, the subjects engage in the testing phase in which they were asked to identify images as “old”, “similar”, or “new” via a labelled button press (192 items total, two seconds each, 0.5 second inter-stimuli interval). The images shown in the second were divided into three categories: exact repetitions of images presented previously (targets), new images not previously seen (foils), and images similar but not identical to images presented in the encoding phase (lures). A new set of images were used for each visit. To correct for any response bias on a per-subject basis pattern separation performance was quantified by calculating the lure discrimination index (LDI). The LDI is the difference between the rate of “similar” responses given to the lure items minus “similar” responses given to foils. Recognition memory performance was assessed using a recognition (REC) score as the difference between the rate of “old” responses given to repeat items minus “old” responses given to foils.

### 2.5. Anthropometry

Weight and body composition were measured using bioelectrical impedance scales (TANITA BC-418 segmental body composition analyzer). Waist circumference and hip circumference were measured using a non-stretch measuring tape around the umbilicus and the widest point over the buttocks, respectively.

### 2.6. Statistics

Statistical analyses were performed using SPSS v25 (IBM Corporation, NY, USA). Characteristics between the IER and CER groups were compared using independent sample t-tests and Mann Whitney U. Between group differences in LDI and REC score were analyzed with one-way ANCOVA controlling for baseline score as a covariate. A paired samples t-test was used to investigate overall changes in LDI score and within-group REC scores. Normal distribution was determined using a combination of the Shapiro-Wilk test of normality and visual analyses of histograms and Q-Q plots. For all analyses, *P* < 0.05 was considered significant.

## 3. Results

### 3.1. Cohort Characteristics

A total of 45 volunteers were enrolled and randomized into the study (Figure 1). However, two participants from the IER group withdrew due to time commitment issues. Therefore, 43 participants successfully completed the study. The CER group had a total of 23 participants consisting of five males and 18 females, and the IER group had a total of four males and 16 females. General characteristics of the participants in each group are detailed in Table 1. There were no significant differences observed at baseline in age, weight, body mass index (BMI), waist circumference, hip circumference, waist to hip ratio, body fat percentage, resting metabolic rate, physical activity levels, and total energy expenditure between groups. 

### 3.2. Compliance and Data Exclusion

Compliance to the intervention was determined by a minimum 2 kg weight loss, 2 cm decrease in waist circumference, or a 500-kcal reduction in reported energy intake. Both groups significantly reduced weight, waist circumference, BMI, body fat %, and resting metabolic rate indicating successful compliance. A total of four subjects were excluded from the data reported here: two participants were excluded for not providing complete dietary data and two others for non-compliance to study protocol.

### 3.3. Energy Restriction, Regardless of Method, Causes Weight Loss and Changes in Body Composition

The endpoint measurements for the CER group was determined as the mean of the data from the follow up visits and for the IER group it was the data collected following two days of not severely restricting energy intake to avoid the acute effects of severe energy restriction. Upon completion of the four-week intervention, both groups saw significant decreases in body composition and measurements. However, these differences were not significant between groups (Table 2).

### 3.4. Energy Restriction, Regardless of Method, Improves Adult-Hippocampal Neurogenesis Dependent Cognition

After adjustment for pre-intervention LDI score, there was no statistically significant difference in post-intervention LDI performance between groups, *F* (1,36) = 0.004, *P* = 0.95). Therefore, in order to investigate the effect of energy restriction regardless of method, on LDI, the groups were treated as a single intervention cohort to analyze the overall change in pattern separation (Figure 2b). There is a statistically significant increase in LDI score (*t* (38) = −3.96, *P* < 0.0005) over four weeks from 0.36 ± 0.25 to 0.46 ± 0.22.

For REC score, after adjustment for the pre-intervention score, there was no statistically significant difference in post-intervention REC performance between groups (*F* (1,36) = 2.464, *P* = 0.125). However, the IER group had a significant reduction in REC score from 0.88 ± 0.10 to 0.77 ± 0.20 (*t* (18) = 3.041, *P* = 0.007), which was not observed in CER (baseline score = 0.84 ± 0.15, endpoint score = 0.82 ± 0.13, *t* (19) = 0.82, *P* = 0.42).

## 4. Discussion

To our knowledge, this is the first study to compare two different methods of energy restriction and their impact on AHN-related cognition in a human population. Our data suggests that there is no significant difference between the two diets in terms of cognitive improvement. However, when analyzed as a single cohort, there was a significant improvement in pattern separation performance. Previous rodent work from other groups have shown energy restriction to be associated with higher survival rates and increased hippocampus-dependent cognition [31,37,38]. Energy restriction can affect the expression of genes involved in neurogenesis and increase neural cell proliferation [39,40]. Furthermore, chronic IER has been associated with a thicker CA1 pyramidal cell layer in the hippocampus proper resulting in enhanced learning and memory [41]. More recently, a study in gerbils showed that cell proliferation in the dentate gyrus was increased by up to 250% with just one month of alternate day IER [42]. We hypothesized that both diets would result in improved performance in the MST, which is being utilized as a form of proxy measure of AHN by testing associated cognition. 

Rodent studies have consistently shown that daily CER can improve many physical and cognitive health factors. For example, 18-month old rats on a 60% calorie restriction diet have a higher survival rate, higher locomotor activity, and a delayed age-related decline in learning and memory [43]. Others have since consistently shown that CER may delay the decline or improve cognitive ability such as spatial discrimination and novel object recognition in rodents [37]. In 2008, it was reported that an energy restriction diet led to the downregulation of genes involved in inflammation and upregulation of neurogenic and developmental genes [44]. Soon after, the same group also showed that CER has an age-dependent effect on the expression of genes involved in stress and also modulates the expression and number of genes involved in Wnt and Notch signal pathways that are involved in development and neurogenesis [39]. Low energy diets during adolescence have been shown to increase cell proliferation and neuron numbers in the hippocampal formation and may result in improved cognition in adulthood [40]. 

IER involves very low-energy intake for varying periods of time, usually 24–48 hours in popular consumer diets. Many religious groups incorporate it into their rituals and is also used in many clinics for weight management, disease prevention, and treatment [45]. Rodent studies have consistently shown a positive impact of nutritional interventions of this nature on AHN. Mice on alternate day feeding schedules have increased survival of newly generated neurons in the granule cell layer of the dentate gyrus [46]. Not only has IER been shown to be able to enhance AHN in healthy mice but also protect neuronal loss caused by excitotoxic damage in the hippocampal formation with increased survival of neurons compared to mice fed ad libitum [47]. Age and disease-related decline in locomotion, exploratory behavior and memory retention are attenuated by IER in mouse models of Alzheimer’s disease further demonstrating neuroprotective benefits [33,48]. Furthermore, ischaemic mice on an IER regime display reduced sensorimotor deficits, and enhanced cell proliferation and survival in the dentate gyrus [49]. In this current study, we were unable to see a difference in cognitive performance between the IER and CER group, suggesting that in humans, it is possible that an overall short-term energy intake reduction is responsible for the improved outcome regardless of the method. In fact, in 2008, Komatsu et al. reported that it is the restriction of energy intake and not diet composition that is important for preventing learning and memory deficits [50]. 

Nevertheless, longer intervention studies are required to determine if there are any significant long-term differences between different methods of energy restriction. However, this must be done with care as prolonged compliance with ER diets can be difficult for participants and could exacerbate pre-existing medical conditions [51,52].

Evidence from human studies looking at the effects of energy restriction on overall health benefits is accumulating. Alternate day feeding in men over a three-year period indicated lower death rates and a 50% reduction in hospital admissions compared to controls on an ad libitum diet [53,54]. A recent human intervention study on strict alternate day fasting showed improvements in markers of general health and healthy ageing with no adverse effects [55]. Several studies on time-restricted feeding, a subcategory of IER involving an eight-hour feeding window during the day, have consistently shown beneficial effects on cardiometabolic health and quality of life [56,57,58]. However, evidence for the impact of energy restriction on the human brain is extremely limited. Obese, post-menopausal women assigned to a 12-week CER intervention significantly improved recognition memory and also displayed increased grey matter volume in the hippocampal formation which could be related to changes in the rate of AHN [59]. Furthermore, trials in healthy men and multiple sclerosis patients have shown that IER can significantly improve depression scores suggesting an ‘antidepressant-like’ effect, potentially through the upregulation of AHN which has been shown to play an important role in mood regulation [60,61]. A ‘fasting-mimicking diet’ developed by Valter Longo resulted in improved markers for ageing, such as IGF-1, after three months. More recently, a study by Jamshed et al. in 2019 showed that time-restricted feeding early in the day tended to increase brain-derived neurotrophic factor as well as have a potential anti-ageing effect by increasing gene expression of longevity genes SIRT1 and MTOR [62]. In the human lifespan, age-related cognitive decline is a natural part of the ageing process, potentially due to the decline in AHN [9,63]. It has been suggested that young new-born neurons within the dentate gyrus may be responsible for mediating pattern separation and as they age mediate the ability to trigger “pattern completion-mediated recall”, i.e., recognition memory [64]. The differential roles dependent on development age of the new-born neurons may explain why pattern separation is age-dependent whereas recognition memory is not [18]. Therefore, it would not be unprecedented to theorize that an intervention which could prevent or slow down the age-related decline in AHN could also prevent naturally occurring age-related impairments.

Unexpectedly, we observed a significant decrease in recognition memory performance in the IER group. This is contrary to what we would expect to happen given the current positive evidence of the beneficial effect of intermittent fasting on health and cognition. This may be explained by the ‘obesity paradox’, whereby a decline in body mass during older age may result in negative cognitive consequences [65,66]. Another study on a sample of adults aged 65 and over found that the overweight subjects performed better in reasoning and visuospatial cognitive testing. However, their memory performance was not significantly better than normal weight subjects, although it is worth noting that it was not worse either [67]. Furthermore, several studies have since published evidence supporting the obesity paradox [68,69]. The cohort used in this study were aged between 35 and 72 years, 12 of whom were over the age of 60. It may be possible that the obesity paradox has come into play in our results.

Despite several limitations of the study there are many strengths to be considered in the design and methods used. Firstly, randomized controlled trials are considered to be the gold standard in human research investigating relationships between an intervention and outcome [70]. Secondly, the food packs provided to the IER group were fortified eliminating the risk of confounding effects of reduced micronutrient intake during fasting. Thirdly, the study had a very low dropout rate (4.4%) with only two participants unable to take part in the study and no adverse effects. Finally, the mnemonic similarity task is highly sensitive to hippocampal function and does not produce a learning effect making it the ideal, clinically-sensitive tool for this study [71].

One of the most important limitations of this study is the fact that there are currently no non-invasive methods of measuring AHN in live humans, only proxy measures, such as those reported here, rendering it difficult to claim that improvements are a direct result of enhanced AHN. Quantification of AHN in humans has only been managed in post-mortem brain samples using immunohistochemical techniques [11,12]. Furthermore, although neuroimaging methods such as fMRI and PET can investigate brain activity and connectivity, they are currently unable to offer a high enough resolution to be able to measure the differentiation and integration of new-born neurons into existing circuits [72]. There is a need for an *in vivo* biomarker for AHN to be measured accurately in living human populations and is currently the biggest obstacle in the field [7]. However, until a validated biomarker has been discovered, proxy measures such as the mnemonic similarity task can be a suitable method to approximate neurogenic activity in humans. Furthermore, it is important to take into considering that this was a short-term four-week intervention, and thus difficult to come to any conclusions on the long-term effects of these diets. Moreover, this study was not designed to investigate cognitive function as a primary outcome and can only provide preliminary evidence that short term energy restriction may improve pattern separation. These findings should be further investigated using a randomized controlled trial powered to test effects on LDI and REC as primary outcomes. Another limitation to consider is that a significantly larger number of females participated in this study compared to males. Men are notoriously difficult to recruit for nutrition related research giving rise to potentially gender-biased results [73]. Therefore, future studies should consider having an equal balance in genders or alternatively be powered to investigate effects in men and women separately. In an ideal scenario, a study such as this would benefit from being carried out in an isolation unit or feeding centre to ensure the intervention is strictly followed per protocol and further work would benefit from being conducted in this way. However, it is also important to consider that removing participants from their usual routine could cause stress, therefore negatively impacting AHN [74]. Moreover, an isolation unit would remove any environmental enrichment which has been shown to have a positive impact on AHN and confound any improvements that result from the interventions [75]. Finally, a non-intervention control group should be included to adjust for confounding factors such as habituation to the research facility environment, which may result in improved concentration in the cognitive test and allow for a more definitive answer as to whether energy restriction is a feasible intervention to improve cognition.

## 5. Conclusions

To the best of our knowledge, this is the first human study to investigate the association between energy restriction, both continuously and intermittently, with neurogenesis-associated cognitive function. In conclusion, energy restriction, regardless of the pattern of restriction, may influence memory function possibly through modulating AHN with the potential to be used as an intervention to prevent or boost cognitive decline. 

## Figures and Tables

**Figure 1 nutrients-12-00638-f001:**
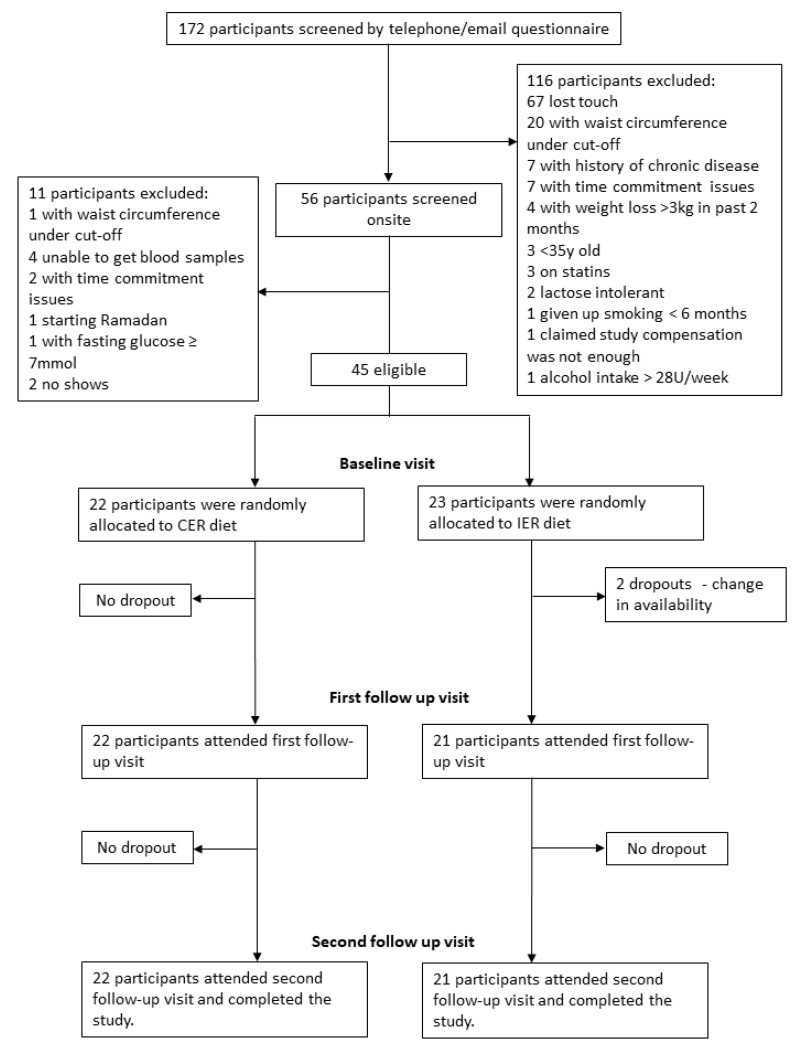
CONSORT (Consolidated Standards of Reporting Trials) diagram for the Met-IER study (Metabolic effects of Intermittent Energy Restriction). IER: intermittent energy restriction; CER: continuous energy restriction.

**Figure 2 nutrients-12-00638-f002:**
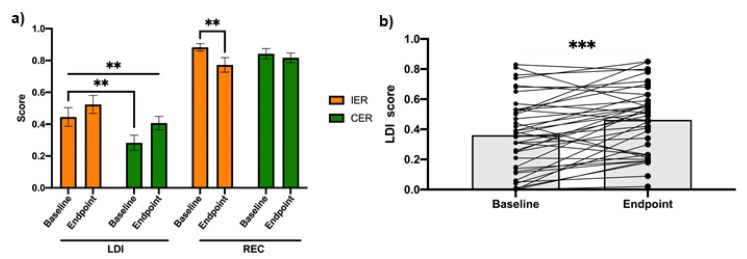
Energy restriction improves pattern separation regardless of method. (**a**) There is no significant difference between IER and CER in pattern separation and recognition memory performance, as measured by LDI and REC scores, respectively. However, the IER group had a significant reduction in REC score (*P* = 0.007), which was not observed in the CER group. (**b**) When the groups are treated as a single cohort, there is a significant increase in LDI score (*P* = 0.0005). Significance at *P* < 0.05. ** *P* < 0.01, *** *P* < 0.001. IER: intermittent energy restriction; CER: continuous energy restriction LDI: lure discrimination index; REC: recognition.

**Table 1 nutrients-12-00638-t001:** Baseline Characteristics. There are no significant differences in baseline characteristics between the intermittent energy restriction (IER) or the continuous energy restriction (CER) group. Data is representative of both genders as a single cohort. Values are presented as mean ± standard deviation. Significance at *P* < 0.05. IER: intermittent energy restriction; CER: continuous energy restriction; BMI: body mass index.

Characteristic	IER (*n* = 20)	CER (*n* = 23)	*P* Value
Age (years)	50.0 ± 12.7	55.7 ± 8.2	0.066
Weight (kg)	87.6 ± 16.8	89.1 ± 20.0	0.820
BMI (kg/m^2^)	32.0 ± 4.7	30.9 ± 5.6	0.529
Waist circumference (cm)	107.7 ± 9.3	110.5 ± 15.8	0.971
Hip circumference (cm)	114.4 ± 8.8	115.7 ± 12.6	0.686
Waist to hip ratio	0.9 ± 0.1	1.0 ± 0.1	0.557
Body fat (%)	39.9 ± 6.5	37.5 ± 7.1	0.258
Resting metabolic rate (kcal/day)	1404.2 ± 276.6	1378.2 ± 304.1	0.772
Physical activity levels	1.44 ± 0.2	1.45 ± 0.1	0.925
Total energy expenditure (kcal/day)	1999.1 ± 357.1	2036.2 ± 476.9	0.818

**Table 2 nutrients-12-00638-t002:** Anthropometric measurements are all decreased after four weeks of energy restriction. There was no significant difference in the percentage change in compliant participants between the groups in any anthropometric measure. Data is representative of both genders as a single cohort. The values are presented as the mean ± standard deviation. Significance at *P* < 0.05. IER: intermittent energy restriction; CER: continuous energy restriction; BMI: body mass index.

Characteristic	IER (*n* = 18)	CER (*n* = 21)	*P* Value
% Change in weight (kg)	−3.1 ± 1.6	−2.8 ± 1.6	0.578
% Change in BMI (kg/m^2^)	−3.1 ± 1.5	−2.8 ± 1.6	0.631
% Change in waist circumference (cm)	−3.8 ± 2.4	−4.3 ± 2.9	0.589
% Change in hip circumference (cm)	−1.9 ± 1.6	−2.0 ± 2.7	0.932
% Change in waist to hip ratio	−1.9 ± 2.5	−2.3 ± 3.1	0.675

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
