# Peer review of "Energy Restriction Enhances Adult Hippocampal Neurogenesis-Associated Memory after Four Weeks in an Adult Human Population with Central Obesity; a Randomized Controlled Trial"

_nutrients, 2020, doi:10.3390/nu12030638_

Round 1
Reviewer 1 Report
I appreciate the opportunity to review this article. I found it to be well written and made a lot of sense.
My major issue with it was the over reliance on acronyms which made it cumbersome to read.
Line 295 -- "feeding" instead of "feeling"
Author Response
- I appreciate the opportunity to review this article. I found it to be well written and made a lot of sense. My major issue with it was the over reliance on acronyms which made it cumbersome to read.
Thank you for your time and comments on our manuscript. We agree that we have overused acronyms and thus to make it easier for the reader we have only kept acronyms that are consistently used throughout the text and removed the following: ER, KCL, VCLD, IPAQ-SF and MST
- Line 295 -- "feeding" instead of "feeling"
Thank you for pointing this out – this has been corrected.
Reviewer 2 Report
Kim et al. has written an interesting article entitled "Energy restriction, regardless of method, enhances
adult hippocampal neurogenesis-associated memory after 4 weeks in an adult human population with central obesity; a randomized controlled trial".
In this study, the author's determined that both continuous and intermittent energy deficiency in the diet is related to neurogenesis and finally in cognitive function improvement.
This study was powered by analyzing 43 human subjects between the age of 35-75 years (both gender) before and after dietary intervention using the mnemonic similarity task as a key technique to understand pattern separation and memory recognition.
Minor comments:
Line: In the title wordings "regardless of method" should be removed.
Line 17-18: The first statement of the abstract is a highly debatable topic.
I suggest the author/s be specific and write that the subventricular zone of the dentate gyrus in the hippocampus is supposed to generate new neurons. If this statement is correct, one should expect that the brain should not have neurodegeneration!
I do not think, there is any problem in the migration of newly form neurons after it's formation.
You should be able to explain this discrepancy if you would like to make a statement that "adult neurogenesis occurs throughout life".
Line 33-34: I do not understand this sentence " ClinicalTrials.gov, NCT02679989. Registered 11 33 February 2016." Please write it in a more complete sentence.
Line 59: Please specify which hippocampal area are known to show pattern generation and recognition memory!
Line 75-76: Could you elaborate where temporary or permanent lesions made specifically in the hippocampal circuit to impair recognition memory?
Line 80: "nutritional cues" may be a better term than that of "environmental cues"
Line 84-88: This sentence could be fitted in the methods section since the "Introduction" section is already long enough.
Methods section:
How one can make sure whether the subject follows IER or CER research intervention? Was there any video monitoring or keeping them in the isolation unit?
Results section:
Why the sampling size is not equal? In both IER and CER, the female population is ~4 times exceeding than males.
Figure 1: Before the baseline visit, there are many variables. Can these variables be also a confounding factor in your results?
Table 1: Please specify if the baseline characteristics refer to both gender in the table Title.
Table 2: Please address it in Table 1.
Major comments/Drawback in the study:
I saw only two figures in your result section, which clearly draws my attention that this study apparently requires further quantification or more results.
If you could add some more analysis that could be an ideal situation. Since you do not examine the subject to fMRI or PET imaging, that could help in this study.
However, you really need to provide some more critical experiments to validate that energy restriction is helping in neurogenesis.
Do you see that whether diet restriction leads the subject to utilize ketone bodies? If that is the case, you might provide such evidence instead of fMRI or PET imaging.
Author Response
Kim et al. has written an interesting article entitled "Energy restriction, regardless of method, enhances adult hippocampal neurogenesis-associated memory after 4 weeks in an adult human population with central obesity; a randomized controlled trial". In this study, the author's determined that both continuous and intermittent energy deficiency in the diet is related to neurogenesis and finally in cognitive function improvement. This study was powered by analyzing 43 human subjects between the age of 35-75 years (both gender) before and after dietary intervention using the mnemonic similarity task as a key technique to understand pattern separation and memory recognition.
We sincerely thank you for your time and insightful comments on our manuscript and have modified it to address your concerns. We believe that your comments have elevated the quality of our manuscript significantly and we hope you will be satisfied with our amendments.
Minor comments:
- Line: In the title wordings "regardless of method" should be removed.
Thank you, this has been removed.
- Line 17-18: The first statement of the abstract is a highly debatable topic.
I suggest the author/s be specific and write that the subventricular zone of the dentate gyrus in the hippocampus is supposed to generate new neurons. If this statement is correct, one should expect that the brain should not have neurodegeneration! I do not think, there is any problem in the migration of newly form neurons after it's formation. You should be able to explain this discrepancy if you would like to make a statement that "adult neurogenesis occurs throughout life".
Thank you for this comment. We absolutely agree that we should be clearer that adult neurogenesis occurs specifically in the subventricular zone of the dentate gyrus and this has been added to line 18. The functional role of adult hippocampal neurogenesis is currently thought to be in mood regulation and brain plasticity behind cognitive processes such as learning and memory (Anacker and Hen, 2017). But there is a lack of evidence at the moment showing involvement in preventing neurodegeneration – only as a response to a brain injury where it has been shown that there can be a small boost to the site (Ngwenya and Danzer, 2019). However, these numbers are too small to have a significant impact. Therefore, indeed as you have stated the low levels of adult hippocampal neurogenesis in humans is not enough to prevent neurodegeneration. Considering the context of this study we do not feel it necessary to include the role in neurodegeneration and brain injury. However, we are very happy to expand the introduction to adult hippocampal neurogenesis if it would be of benefit to the reader.
Anacker, C. and Hen, R. (2017) ‘Adult hippocampal neurogenesis and cognitive flexibility-linking memory and mood’, Nature Reviews Neuroscience. Nature Publishing Group, pp. 335–346. doi: 10.1038/nrn.2017.45.
Ngwenya, L. B. and Danzer, S. C. (2019) ‘Impact of traumatic brain injury on neurogenesis’, Frontiers in Neuroscience. Frontiers Media S.A., 13(JAN). doi: 10.3389/fnins.2018.01014.
- Line 33-34: I do not understand this sentence " ClinicalTrials.gov, NCT02679989. Registered 11 33 February 2016." Please write it in a more complete sentence.
We have rewritten this declaration as a full sentence to be clearer as follows: “This study was registered on ClinicalTrials.gov, (NCT02679989) on 11 February 2016.” (line 34)
- Line 59: Please specify which hippocampal area are known to show pattern generation and recognition memory!
Thank you for pointing this out, it is indeed important that we make sure to distinguish the differential roles of the hippocampal areas. We have specified that the dorsal hippocampal formation is associated with pattern separation and recognition memory and included the following reference (line 62): Fanselow MS, Dong HW. Are the Dorsal and Ventral Hippocampus Functionally Distinct Structures? Vol. 65, Neuron. NIH Public Access; 2010. p. 7–19.
- Line 75-76: Could you elaborate where temporary or permanent lesions made specifically in the hippocampal circuit to impair recognition memory?
We completely agree that ideally it would be great if we could include this information. The rodent paper is a systematic review that includes studies that do specify the dorsal hippocampus and others that just call it the hippocampus as a whole. Similarly, for the primate study the lesions were created throughout the entire hippocampal formation. Therefore, we have elaborated on this citation to include this information (line 78).
- Line 80: "nutritional cues" may be a better term than that of "environmental cues"
Thank you, we agree that “nutritional cues” makes much more sense. This has been corrected in line 82.
- Line 84-88: This sentence could be fitted in the methods section since the "Introduction" section is already long enough.
Thank you for your suggestion and although we do agree that this sentence would fit well in the methods we believe that this sentence is better suited in the introduction as it is introducing the different methods of energy restriction that are used in the majority of animal and human studies. Moreover, it is a supporting statement introducing the two different energy restriction interventions that are used in our study. The delivery of the interventions is then explained in detail in the methods later in section 2.3, line 143-146.
Methods section:
- How one can make sure whether the subject follows IER or CER research intervention? Was there any video monitoring or keeping them in the isolation unit?
We agree that in an ideal scenario an isolation unit would have been the ideal method to ensure that the intervention is followed properly. Unfortunately, these resources weren’t available to us when this study was conducted. However, in order to maximise compliance all participants received follow up phone calls and attended group support sessions (stated in section 2.3). Furthermore, we determined compliance to the intervention by a minimum 2kg weight loss, 2cm decrease in waist circumference or a 500-calorie reduction in reported energy intake (stated in section 3.2). It would absolutely be very beneficial for further work to be conducted in such a way that the participants can be monitored better and is something we will consider for future studies. However, we also believe that due to the fact that we are looking at whether an intervention can improve cognition in our cohort, removing them from their usual routine and therefore removing any environmental enrichment which has been shown to be beneficial for neurogenesis could confound any improvements that result from the intermittent fasting. We have included this in the discussion (350-356).
Results section:
- Why the sampling size is not equal? In both IER and CER, the female population is ~4 times exceeding than males.
We agree that it would be more desirable to have a better balance of genders in our study but unfortunately this was the population that was able to be recruited. Men are notoriously difficult to recruit for nutrition related research (Rounds and Harvey, 2019). Therefore, to tackle this issue in future studies we will consider recruiting only one gender to eliminate any potential gender-biased results. We have ensured to mention this in the discussion (line 346-350).
Rounds, T. and Harvey, J. (2019) ‘Enrollment Challenges: Recruiting Men to Weight Loss Interventions’, American Journal of Men’s Health. SAGE Publications Inc., 13(1). doi: 10.1177/1557988319832120.
- Figure 1: Before the baseline visit, there are many variables. Can these variables be also a confounding factor in your results?
The variables stated in Figure 1 are reasons that participants were excluded from the study. Indeed, these would have been confounding factors in our results. The inclusion and exclusion criteria stated in section “2.2 Participant Selection” ensures that there were no participants in the study with baseline variables that are likely to confound our results.
- Table 1: Please specify if the baseline characteristics refer to both gender in the table Title.
- Table 2: Please address it in Table 1.
Both figure legends have been amended to address this.
Major comments/Drawback in the study:
- I saw only two figures in your result section, which clearly draws my attention that this study apparently requires further quantification or more results. If you could add some more analysis that could be an ideal situation. Since you do not examine the subject to fMRI or PET imaging, that could help in this study. However, you really need to provide some more critical experiments to validate that energy restriction is helping in neurogenesis. Do you see that whether diet restriction leads the subject to utilize ketone bodies? If that is the case, you might provide such evidence instead of fMRI or PET imaging.
Thank you for your insightful feedback and we do agree it would be ideal to be able to add further data proving a direct impact of the interventions on neurogenesis. Unfortunately, there are currently no non-invasive methods of quantifying neurogenesis in living humans hence why we are relying on the use of hippocampus-associated measures such as the mnemonic similarity task. Any quantification of neurogenesis in humans has only been managed in post-mortem brain samples using immunohistochemical techniques. Furthermore, although fMRI and PET imaging are useful to investigate brain activity and connectivity, they do not have high enough resolution for us to be able to measure adult neurogenesis, i.e. the differentiation and integration of new-born neurons in existing circuits. There is no question that there needs to be an in vivo biomarker for neurogenesis for us to be able to measure it accurately in living human populations and it is the biggest obstacle in the field. But until a validated biomarker has been discovered, we believe that proxy measures such as this are a valid method to approximate neurogenic activity in humans. (Ho et al., 2013; Lee and Thuret, 2018). This has been included in the discussion (line 333-340).
Ho, N. F. et al. (2013) ‘In vivo imaging of adult human hippocampal neurogenesis: progress, pitfalls and promise’, Molecular Psychiatry. Nature Publishing Group, 18(4), pp. 404–416. doi: 10.1038/mp.2013.8.
Lee, H. and Thuret, S. (2018) ‘Adult Human Hippocampal Neurogenesis: Controversy and Evidence’, Trends in Molecular Medicine. Elsevier Ltd, pp. 521–522. doi: 10.1016/j.molmed.2018.04.002.
Reviewer 3 Report
This is a well-executed and comprehensive study documenting the association between energy restriction with neurogenesis-associated cognitive function. The aim of the study is clearly significant and the description of findings is accurate. I believe that only minor revisions concerning the anatomical nomenclature are needed. In fact, the human hippocampal region consists of two sets of cortical structures, the hippocampal formation and the parahippocampal region (see Insausti and Amaral, Hippocampal formation in: The Human Nervous System, Mai Paxinos, 2012). The hippocampal formation is composed of the dentate gyrus, the hippocampus proper (which is subdivided into three fields: CA3, CA2 and CA1) and the subiculum (Scharfman et al., 2000; Witter and Amaral, 2004; Furtak et al., 2007). The parahippocampal region includes the presubiculum, the parasubiculum, the entorhinal cortex, the perirhinal cortex and the parahippocampal cortex (Scharfman et al., 2000; Witter and Amaral, 2004; Furtak et al., 2007). In this article the authors use the generic term hippocampus, but from a terminological point of view it is not correct. Consequently:
- page 1, line 18; page 2 , line 47; page 2, line 65, page 2, line 74; page 2, line 75; page 7, line 259; page 7, line 266; page 8 line 289: use "hippocampal formation" instead of "hippocampus";
- page 7, line 244: use "hippocampus proper" instead of "hippocampus".
Author Response
This is a well-executed and comprehensive study documenting the association between energy restriction with neurogenesis-associated cognitive function. The aim of the study is clearly significant and the description of findings is accurate. I believe that only minor revisions concerning the anatomical nomenclature are needed. In fact, the human hippocampal region consists of two sets of cortical structures, the hippocampal formation and the parahippocampal region (see Insausti and Amaral, Hippocampal formation in: The Human Nervous System, Mai Paxinos, 2012). The hippocampal formation is composed of the dentate gyrus, the hippocampus proper (which is subdivided into three fields: CA3, CA2 and CA1) and the subiculum (Scharfman et al., 2000; Witter and Amaral, 2004; Furtak et al., 2007). The parahippocampal region includes the presubiculum, the parasubiculum, the entorhinal cortex, the perirhinal cortex and the parahippocampal cortex (Scharfman et al., 2000; Witter and Amaral, 2004; Furtak et al., 2007). In this article the authors use the generic term hippocampus, but from a terminological point of view it is not correct. Consequently:
- page 1, line 18; page 2 , line 47; page 2, line 65, page 2, line 74; page 2, line 75; page 7, line 259; page 7, line 266; page 8 line 289: use "hippocampal formation" instead of "hippocampus";
- page 7, line 244: use "hippocampus proper" instead of "hippocampus".
Firstly, we’d like to thank you for your time reading and reviewing our manuscript. We agree that this is an important distinction to make that we have missed. All the corrections above have been made.
Round 2
Reviewer 2 Report
I have no further comments, the author/s have addressed all my queries.
I come to understand that due to the lack of adequate tools and techniques to probe into the human brain directly with non-invasive methods, the dimension of this study was obviously confined.
However, those limitations I pointed still remain one of the great challenges in the filed. Now, the authors have adequately explained those setbacks in their manuscript, I am convinced that more experiments seems very impossible unless proper tools and techniques are developed to probe the human adult neurogenesis directly!